# Potential of *Hibiscus sabdariffa* L. and Hibiscus Acid to Reverse Skin Aging

**DOI:** 10.3390/molecules27186076

**Published:** 2022-09-17

**Authors:** Duanyang Wang, Maki Nagata, Masako Matsumoto, Yhiya Amen, Dongmei Wang, Kuniyoshi Shimizu

**Affiliations:** 1Department of Agro-Environmental Sciences, Graduate School of Bioresource and Bioenvironmental Sciences, Kyushu University, 744 Motooka, Nishi-ku, Fukuoka 819-0395, Japan; 2Department of Pharmacognosy, Faculty of Pharmacy, Mansoura University, Mansoura 35516, Egypt

**Keywords:** *Hibiscus sabdariffa* L., bioactive compound, skin fibroblasts, aging, oxidative stress, anti-inflammatory, bio-guided isolation

## Abstract

*Hibiscus sabdariffa* L. (*HS*) has a long history of edible and medicinal uses. In this study, the biological activities of the extracts, chromatographic fractions, and hibiscus acid obtained from *HS* were evaluated for their potential bioactivities. Their ability to promote extracellular matrix synthesis in skin fibroblasts was evaluated by enzyme-linked immunosorbent assays. Their anti-inflammatory activity was evaluated in a nitric oxide (NO)–Griess inflammatory experiment. Furthermore, hibiscus acid was found to have a strong anti-oxidative stress effect through the establishment of an oxidative stress model induced by hydrogen peroxide. Several assays indicated that hibiscus acid treatment can effectively reduce extracellular adenosine triphosphate (ATP) secretion and carbonyl protein production, as well as maintain a high level of reduced/oxidized glutathione (GSH/GSSG) in skin cells, thus providing a possible mechanism by which hibiscus acid can counter antioxidative stress. The present study is the first to explore the reversing skin aging potential and the contributory component of *HS*.

## 1. Introduction

*Hibiscus sabdariffa* L. (*HS*), also known as roselle, is a wild tropical plant belonging to the Malvaceae family [1]. It has noticeable red calyces, originates in Asia (India to Malaysia) and has been cultivated all over the world as a crop of considerable value. *HS* has a long history of edible and medicinal uses spanning from Egypt, China, Thailand, and Indonesia to South America [1,2,3,4,5]. There have been many studies of the pharmacological behaviors and health attributes of *HS*. *HS* calyx has been studied widely with regard to its blood pressure-lowering, anti-obesity, anti-anaemia, antimicrobial and natriuretic effects [6,7,8]. As demonstrated in a number of in vitro and in vivo studies, *HS* is very useful both as food and as a pharmaceutical component. However, few studies have focused on the application of *HS* in treating skin aging.

Skin is regarded as the largest organ of the human body, as well as the boundary between an organism and its environment [9]. As people pay increasing attention to skin care, research on skin care treatments, especially anti-aging treatments, has become more prevalent [10,11,12]. Skin aging, like the aging of other organs, is characterized by the progressive loss of functionality and regenerative potential [13]. The morphology and structure of human skin and its physiological functions show degenerative changes that occur with age. These changes can be divided into exogenous aging and endogenous aging [14,15]. Exogenous aging is mainly due to exposure to ultraviolet light, environmental pollution and unhealthy habits, such as smoking and drinking. This leads to skin wrinkles, discoloration and skin relaxation. Endogenous aging is driven by endocrine factors and genetic factors and the skin’s natural tendency to atrophy and develop dry wrinkles.

Located in the dermis of the skin, normal human dermal fibroblasts (NHDFs) are the cells that work with the extracellular matrix to secrete elastin, collagen, cytokines, etc., all of which play vitally important roles in reversing the process of skin aging. As the number of fibroblasts decreases, however, the capacity for proliferation declines, and the metabolic function deteriorates during aging [16]. Therefore, the collagen and hyaluronic acid contents in fibroblasts are considered to be the critical indicators of anti-aging performance. Apart from various internal factors, skin aging caused by external factors involves the destruction of the skin barrier, the occurrence of an oxidative stress reaction and reactive oxygen species [17,18], as well as the inhibition of immune function and inflammatory stress and the consequent production of inflammatory factors [19,20]. Therefore, antioxidative stress and the anti-inflammatory effect are regarded as important evaluation methods of anti-aging performance.

## 2. Results

### 2.1. HS Has Prominent Potential for Reversing Skin Aging without Cytotoxicity

In order to screen the potential bioactivities of *HS* calyx, as well as select the appropriate extraction method, the dried calyces of *HS* was extracted with water and ethanol to obtain *HS* aqueous extract (HSA) and *HS* ethanol extract (HSE), respectively. Their safety concentration ranges were first detected. As the MTT results show in Figure 1A, neither HSA nor HSE showed toxicity to skin fibroblasts, even at high concentrations up to 1000 μg/mL. 

The results shown in Figure 1B indicate that treatment with HSA and HSE significantly reduced the production of NO in RAW 264.7 cells without cytotoxicity (Appendix A), thus reducing the inflammatory response. According to the results in Figure 1C,D, HSE significantly stimulated the production of collagen and hyaluronic acid in NHDF-Ad. The 500 μg/mL concentrations of HSE promoted the secretion of collagen and hyaluronic acid by 48.1% and 24.3%, respectively.

According to the above evaluation of the biological activity of *HS* crude extracts, the ethanol extract HSE significantly promoted the synthesis of collagen and hyaluronic acid in fibroblasts, and HSA and HSE showed significant anti-inflammatory activity. Thus, it can be concluded that *HS* has low toxicity and may have excellent potential to reverse skin aging. Moreover, HSE was selected as the object for bio-activity-guided fractionation and isolation.

### 2.2. Activity Verification and Characterization of Bio-Guided Fractionation

The fractionation of 18 g HSE was carried out using the column chromatographic technique. The quantity and yield of each fraction are shown in Table 1. In order to determine the active fraction from the total extract, the proportion of each fraction in the total extract should be considered. Thus, in the subsequent bioactivity experiments, the concentration of each fraction was determined from its proportion in the total extract. 

In the previous experimental results, 500 μg/mL of HSE showed appropriate activity, so the concentration of each fraction was calculated for those obtained using 500 μg/mL HSE. The yield ratios of the 10 fractions were 0.35%, 0.69%, 0.17%, 0.54%, 8.28%, 37.76%, 8.27%, 10.79%, 14.36% and 16.14%, respectively. Therefore, the corresponding concentrations of the 10 fractions obtained using 500 μg/mL HSE were 1.73 μg/mL, 3.45 μg/mL, 0.86 μg/mL, 2.68 μg/mL, 41.39 μg/mL, 188.78 μg/mL, 41.33 μg/mL, 53.94 μg/mL, 71.79 μg/mL and 80.67 μg/mL, respectively.

The results shown in Figure 2A indicate that Fr. 5 and Fr. 6 had significant stimulus effects on the production of collagen and hyaluronic acid in NHDF-Ad. They also showed an extremely significant inhibitory effect on the production of NO in RAW 264.7 cells under the inflammatory response. None of these tests showed cytotoxic effects (Appendix A). Figure 2B shows the HPLC-ELSD chromatograms for Fr. 5 and Fr. 6. These results indicate that the two fractions are very similar in chemical composition, with one component present in large amounts in both fractions (LC/MS-qTOF total ion chromatogram spectrum of HSE, Fr. 5 and Fr. 6, provided in the Appendix A). In particular, Fr. 6, which had the most significant activity, contained almost only this one ingredient. Therefore, the isolation was conducted using Fr. 6. The bio-activity-guided fractionation and isolation steps are shown in Figure 2C.

### 2.3. Characterization of Hibiscus Acid and H_2_O_2_-Induced Oxidative Stress Model

Fr. 6 was separated by column chromatography and compound I was obtained. From the NMR and MS detection data and comparisons with previously reported data [8,21], compound I was identified as hibiscus acid (Figure 3A). The identification information was as follows: (2*S*,3*R*)-3-hydroxy-5-oxotetrahydrofuran-2,3-dicarboxylic acid. $\left[\alpha \right]\begin{array}{c} 20\\ D \end{array}\;$= +111 (c 0.58, H_2_O); δ_H_ (acetone-d6) 5.35 (1H, s, H-2); 3.25 (1H, d, *J* = 17.20 Hz, H-4a); 2.78 (1H, d, *J* = 17.20 Hz, H-4b). δ_C_ (acetone-d6) 173.05, 166.99, 83.04, 78.58, 42.20; HR-ESI-MS (negative) m/z 189.0055 [M-H]-, 379.0158 [2M-H]-. 

As shown in Figure 3C,D, hibiscus acid showed a highly stimulating effect on hyaluronic acid and a moderately stimulating effect on collagen in NHDF-Ad, without cytotoxicity even at a high concentration (Figure 3B). It also reduced the production of NO in RAW 264.7 cells in an LPS-induced inflammation response. In addition, considering the stability, HPLC analysis indicated that hibiscus acid remained stable in cell experiments (Appendix A).

In order to further understand the mechanism by which *HS* and hibiscus acid reverse skin aging, hydrogen peroxide (H_2_O_2_) was used to establish a cellular aging model in NHDF-Ad. As shown in Figure 3F, NHDF-Ad cells were treated with different concentrations of H_2_O_2_ for different durations. As the H_2_O_2_ concentration increased, the cell death rate gradually increased. Figure 3G presents electron microscopy images showing the cell changes induced by the H_2_O_2_ treatment. At the same time, as shown in Figure 3H, the cell viability decreased by around 50% after treatment with 0.8 mM H_2_O_2_, while pretreatment with hibiscus acid significantly improved the cell viability. These results provide further evidence that hibiscus acid pretreatment could effectively induce resistance to the cell damage induced by oxidative stress.

### 2.4. Evaluation of the Possible Mechanisms of Hibiscus Acid on Anti-Oxidative Stress Activities

In order to further evaluate the mechanism by which hibiscus acid reverses skin aging, several different experimental approaches were followed. Considering the structure of human skin, epidermal cells were used as a new model for the evaluation method. Studies have shown that extracellular ATP can be used as a messenger molecule to regulate the cell metabolism through specific signaling mechanisms [22]. An increase in a certain level of extracellular ATP can stimulate the increase of intracellular free calcium ions, NO, reactive oxygen species and other second messenger molecules, thereby affecting the cell metabolism and triggering cell senescence and death [23,24]. Protein carbonylation is a non-enzymatic and irreversible carbonyl modification of proteins that mainly involves the formation of protein carbonyl compounds through a stress reaction with reactive carbonyl species from a different source [25]. In cells, glutathione exists in reduced (GSH) and oxidized (GSSG) states and acts as an oxidant to protect cells from free radical damage [26,27]. The GSH/GSSG ratio can reflect this imbalance.

According to the results shown in Figure 4B, the extracellular ATP production of HaCaT cells was significantly downregulated after hibiscus acid treatment without the effect of hibiscus acid itself (Appendix A). A hibiscus acid concentration of 1 μg/mL already showed a significant effect, as it reduced ATP production by 30.4%. As shown in Figure 4C, the carboxyl protein content of HaCaT cells was significantly downregulated after the hibiscus acid treatment. Furthermore, according to Figure 4D, the ratio of GSH/GSSG in HaCaT cells was significantly upregulated after the hibiscus acid treatment. Thus, the mechanism by which hibiscus acid reverses skin aging could be one of the following: (1) hibiscus acid reduces the damage to protein production by affecting the oxidative binding between free radicals and proteins in cells; (2) hibiscus acid inhibits the transformation from GSH to GSSG in cells, upregulates the intracellular GSH/GSSG ratio and maintains a highly reductive intracellular environment; and (3) hibiscus acid reduces the release of ATP from the cells to outside the cells, further reducing the secretion of various aging-related phenotypes in skin cells and delaying the aging process of skin cells (Figure 4A).

## 3. Discussion

The aging population has significantly increased globally due to improvements in living standards and advances in health technology. Aging has become a major global concern due to increases in the number of the elderly. Aging is a stress response induced by multiple intrinsic and extrinsic insults, including oxidative stress, inflammatory stress, radiation stress, genotoxic stress, mitochondrial dysfunction and oncogenic activation. Several studies have recently reported that cellular aging is associated with body aging [28,29,30]. Aging cells continuously accumulate during the body aging process. Elimination of aging cells delays and reverses the aging process in animal models [31,32,33,34,35,36,37]. Skin serves as the first line of defense when the body comes into contact with the external environment. However, as we age, our skin becomes more susceptible to external stimuli and invasion. In addition to the natural aging caused by age, air pollutants, UV radiation, tobacco smoke, chemicals and toxic skin care products can trigger stress responses in skin cells, thereby exacerbating skin aging. Aging of the skin is associated with inflammation, disruption of the skin barrier, disruption of extracellular matrix, reduced repair capacity and an increased risk of skin cancer [38]. Although we cannot stop the aging process, interventional studies indicate that the use of appropriate nutritional supplements can delay skin aging and enhance skin condition. Two main types of compounds are used as ingredients in anti-aging reagents. The first is antioxidants, such as flavonoids, polyphenols and vitamins. The second is cell regulators, such as vitamin A derivatives, peptides and botanicals, which directly affect skin collagen metabolism or regulate the secretion and synthesis of collagen [39].

*HS* has relatively easy growth and is a common crop in many countries. It is mainly used as food. In addition, it is used in traditional medicine, as a source of fiber and animal feed and in the cosmetics industry. However, findings on its cosmetic use are currently only available in Malaysia, where it is used for the production of scrubs and soaps [4]. Previous research on *HS* mainly focused on the anti-obesity, anti-diabetic, and blood pressure- and blood lipid-lowering effects of the plant. However, its potential as a cosmetic skin care product has not been fully elucidated. Several in vitro and in vivo studies have demonstrated the potent antioxidant effects of *HS* extracts [40,41]. For instance, a previous in vivo study reported that *HS* extract shows anti-inflammatory activity and that the ethanol extract was better than the water extract, which is consistent with the findings of the present study [42]. In addition, some researchers have explored its antibacterial activity [43,44]. In the current study, we conducted various screening experiments to explore the skin care potential of *HS*. The findings indicated that *HS* has great potential for the development of skin anti-aging products.

In conclusion, the active components and potential activity of *HS* from Okinawa, Japan, were explored in this study. To the best of our knowledge, the crude extract, chromatographic fractions and isolated compounds of *HS* were used to evaluate the anti-aging effect and to determine the major active components for the first time in this study. The study findings provide a new approach and a new scientific basis for the development of *HS* and its utilization in delaying skin aging. In addition, the *HS* crude extract seems to have stronger activity compared with hibiscus acid alone. This result indicates that the activity of *HS* is likely a result of synergistic effects from more than one compound. Further studies should be conducted to explore the specific mechanism of *HS* in reversing skin aging.

## 4. Materials and Methods

### 4.1. Chemicals and Instruments

Ethanol (Wako), acetonitrile (Wako), methanol (Wako), sulfuric acid (Wako), acetone (Wako), trifluoroacetate (Tokyo Chemical Industry), dimethyl sulfoxide (Wako), MTT (3-(4,5-dimethyl-2-thiazolyl)-2,5-diphenyltetrazolium bromide), LPS and Griess reagent were all purchased from Sigma (St. Louis, MO, USA). An ELx 800 Universal Microplate Reader (BIO-TEK), VD-250R Freeze Dryer (TAITEC), US-105 Sonicator (SND), 5420 Centrifuge (IMOTO), R-300 Rotavapor (BUCHI), V-300 Vacuum Pump (BUCHI), High Performance Flash Chromatography (HPFC) system (Biotage AB), Medium Pressure Liquid Chromatography (MPLC) system (EPCLC, Yamazen), Preparative High Performance Liquid Chromatography (PHPLC) system (EPCLC, Yamazen), 1220 Infinity LC (Agilent Technologies), NMR spectrometer (Bruker DRX-600; Bruker Daltonics, Billerica MA, USA), Quadrupole time-of-flight (qTOF) mass spectrometer (Agilent Technologies, USA) and JASCO DIP-370 polarimeter (JASCO, Tokyo, Japan) were used.

### 4.2. Plant Samples, Extraction and Fractionation

The dried calyces of *HS* were purchased from NAKAZEN Co., Ltd. in April 2019. The dried *HS* (200 g) was extracted with ethanol by maceration (1 h ultrasound per day) over 72 h at room temperature. The extracts were then filtered and rotary evaporated to dryness under reduced pressure. The extraction yield of HSE was 15.3%. The aqueous extraction procedure repeated the same steps, but the solvent was changed to MilliQ. The extraction yield of HSA was 20.4%. The HSE was subjected to silica gel column chromatography using an HPFC system and eluted with an n-hexane-EtOAc gradient (100:0 → 0:100) to an EtOAc-MeOH gradient (100:0 → 0:100). Similar fractions were pooled together based on TLC and high-performance liquid chromatography (HPLC) analysis. In the end, 10 fractions were obtained, and all fractions were further tested for their bioactivity.

### 4.3. Isolation and Identification

Fraction 6 was first purified in an MPLC system and, secondly, in a preparative HPLC system. In both cases, the elutes were analyzed using a 1220 Infinity LC. The fraction was eluted with an H_2_O-MeOH gradient (5:95 → 0:100) in the MPLC system and afforded seven sub-fractions. The sub-fraction Fr. 6-1 contained the most major compounds; hence, it was eluted with H_2_O-MeOH (5:95) by preparative HPLC repeatedly. Finally, compound **1** was isolated and purified. The isolated compound was identified by performing NMR spectroscopy, mass spectrometry and polarimetry analyses.

### 4.4. Cell Culture and Cytotoxicity Experiment

Adult normal human dermal fibroblasts (NHDF-Ad), human keratinocytes (HaCaT) and mouse macrophage RAW 264.7 cells were obtained from Riken Bio-resource Center Cell Bank (Tsukuba, Japan). The cells were cultured in Dulbecco’s Modified Eagle Medium (DMEM) supplemented with high glucose, 10% fetal bovine serum (FBS) and 1% penicillin and streptomycin at 37 °C in a 5% CO_2_ humidified incubator. Cell viability was assessed using a modified MTT assay. The optical density of each well was measured at 570 nm with a microplate reader.

### 4.5. Anti-Inflammatory Experiment

RAW 264.7 cells were seeded on 96-well plates for 24 h before the treatment. The medium was replaced with a mixture of sample solution and LPS (1 μg/mL) in DMEM. After 24 h cultivation, the NO concentration in the cultured medium was determined via the Griess reaction. The optical density was determined at 570 nm with a microplate reader.

### 4.6. Collagen and Hyaluronic Acid Production Assay

NHDF-Ad cells were routinely maintained in 10% FBS and 1% anti-biotic-antimycotic in DMEM. After the cell seeding and sample treatment, the amount of collagen and hyaluronic acid in the supernatant was measured using a human type I collagen enzyme-linked immunosorbent assay (ELISA) kit (ACEL, Kanagawa, Japan) and an QnE hyaluronic acid ELISA kit (Biotech Trading Partners, Encinitas, CA, USA), separately.

### 4.7. H_2_O_2_-Induced Intracellular Oxidative Stress Model

In order to investigate the survival time and concentration of half of the human skin fibroblasts damaged by H_2_O_2_, NHDF-Ad cells were prepared in 96-well plates and treated with different concentrations of H_2_O_2_ (0.2, 0.4, 0.6, 0.8, 1.0 and 1.2 mM) for different treatment times (1, 3, 6 and 12 h). Then, the effects of different concentrations of H_2_O_2_ on the activity of the human skin fibroblasts were detected with a conventional MTT assay.

### 4.8. Extracellular ATP, Protein Carbonyl and GSSG/GSH Quantification Assays 

The extracellular ATP content, intracellular protein carbonyl content and GSSG/GSH ratio were assessed with an ATP assay kit (Toyo B-net Co., Ltd., Tokyo, Japan), a protein carbonyl content assay kit (BioVision) and a GSSG/GSH quantification kit (Dojindo Molecular Technologies, Inc), according to the manufacturers’ instructions.

### 4.9. Statistical Analysis 

Results are shown as the mean values ± standard deviation (SD) of at least three independent experiments (n = 3). *p*-values less than 0.05 were considered to be statistically significant.

## Figures and Tables

**Figure 1 molecules-27-06076-f001:**
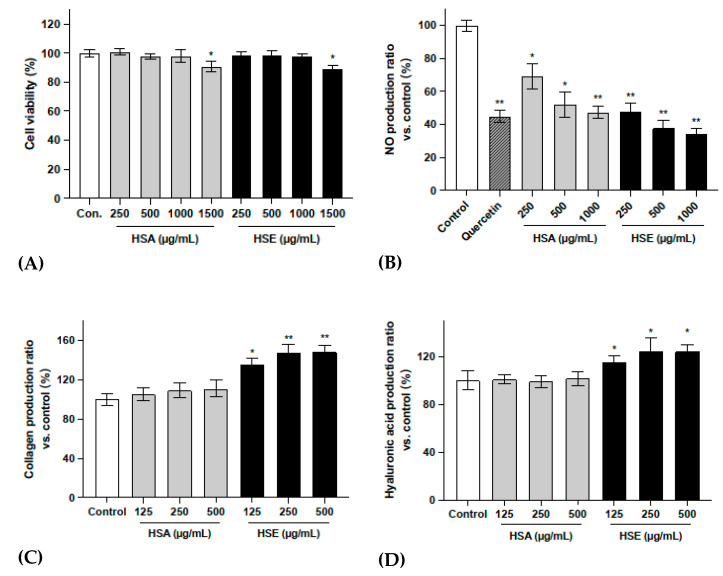
Skin aging-related bioassay results for *HS* crude extract. (**A**) Cytotoxicity of *HS* crude extract to skin fibroblasts. NHDF-Ad seeded in a 96-well micro-plate. After treatment with different concentrations of HSA and HSE, the cell viability was measured by MTT assay. (**B**) The NO production ratio in RAW 264.7 cells after LPS stimulation and treatment with different concentrations of HSA and HSE. Quercetin with concentration of 10 μM. The collagen (**C**) and hyaluronic acid (**D**) production ratio in NHDF-Ad after treatment with different concentrations of HSA and HSE. All data were subjected to multiple comparative analysis by two-way ANOVA, followed by Dunnett’s post hoc test as appropriate. * *p* < 0.05; ** *p* < 0.01 compared with the control group.

**Figure 2 molecules-27-06076-f002:**
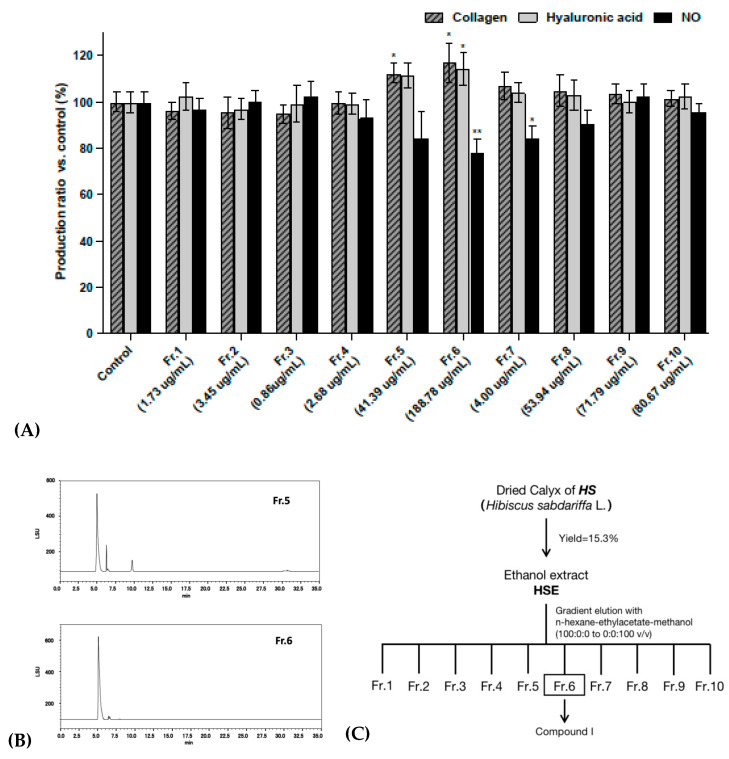
(**A**) Collagen and hyaluronic acid production in NHDF-Ad treated with 10 fractions measured by ELISA kit. Anti-inflammatory activity was tested by LPS-induced NO production in RAW 264.7 cells. (**B**) HPLC-ELSD chromatograms of Fr. 5 and Fr. 6. (**C**) Scheme illustrating the bioactivity-guided fractionation procedure for *HS*. All data were subjected to multiple comparative analysis by two-way ANOVA, followed by Dunnett’s post hoc test as appropriate. * *p* < 0.05; ** *p* < 0.01 compared with the control group.

**Figure 3 molecules-27-06076-f003:**
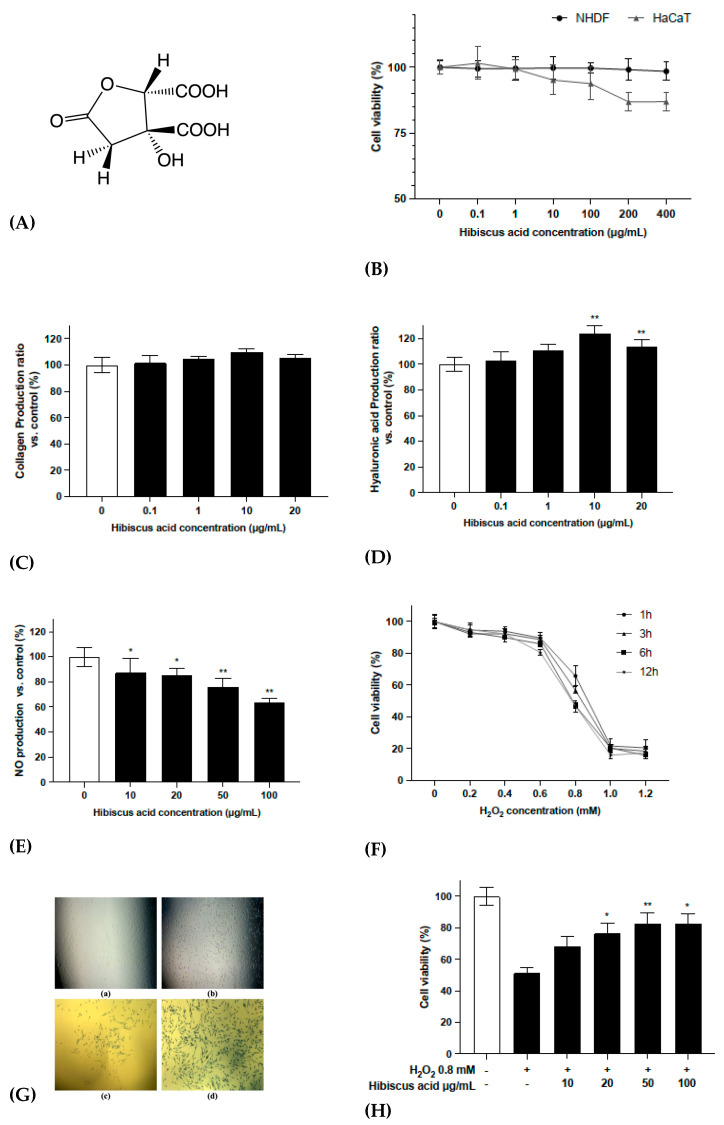
(**A**) Structure of hibiscus acid. (**B**) The cytotoxicity of hibiscus acid toward NHDF-Ad and HaCaT cells. (**C**,**D**) The production of collagen and hyaluronic acid in hibiscus acid-treated NHDF-Ad. (**E**) The NO production of hibiscus acid-treated RAW 264.7 under LPS-induced inflammatory stress. (**F**) Model of intracellular oxidative stress induced by H_2_O_2_. NHDF-Ad was seeded to 96-well microplates, treated with different concentrations of H_2_O_2_ and treatment times and the final cell viability was detected with the MTT method. (**G**) The morphology of fibroblasts under different conditions photographed under a microscope. (a) The NHDF-Ad under normal conditions. (b) NHDF-Ad with 0.8 mM H_2_O_2_, 3 h. (c) NHDF-Ad with H_2_O_2_, 3 h, after MTT processing. (d) NHDF-Ad pretreatment with hibiscus acid and H_2_O_2_, 3 h, after MTT processing. (**H**) The protective effect of hibiscus acid in H_2_O_2_-induced oxidative stress model of NHDF-Ad. All data were subjected to multiple comparative analysis by two-way ANOVA, followed by Dunnett’s post hoc test as appropriate. * *p* < 0.05; ** *p* < 0.01 compared with the “0” (control) group.

**Figure 4 molecules-27-06076-f004:**
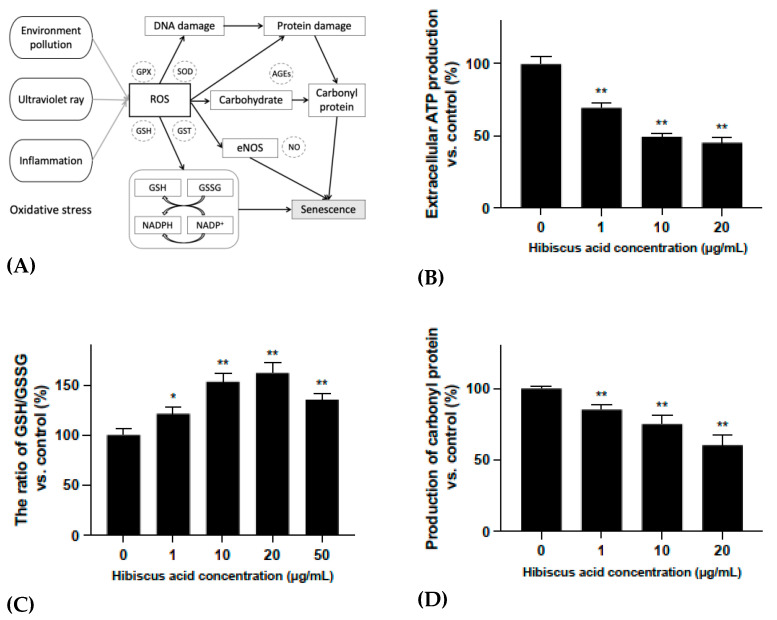
(**A**) The possible mechanism of intracellular oxidative stress. (**B**) The ratio of extracellular ATP production of hibiscus acid-treated HaCaT cells to that of cells without hibiscus acid treatment (control). (**C**) The ratio of carbonyl production of hibiscus acid-treated HaCaT cells to that of cells without hibiscus acid treatment (control). (**D**) The ratio of GSH/GSSG production of hibiscus acid-treated HaCaT cells to that of cells without hibiscus acid treatment (control). All data were subjected to multiple comparative analysis by two-way ANOVA, followed by Dunnett’s post hoc test as appropriate. * *p* < 0.05; ** *p* < 0.01 compared with the “0” (control) group.

**Table 1 molecules-27-06076-t001:** Ten fraction collections from 18 g *HS* calyx EtOH extract classified according to thin-layer chromatography. The yield ratio was calculated by dividing the yield (mg) by 18 g of HSE. The final total yield reached 97.32%. Assay concentration means the concentration used for the following bio-assays, and it was calculated with the yield ratio of each fraction corresponding to 500 μg/mL HSE.

	Fr. 1	Fr. 2	Fr. 3	Fr. 4	Fr. 5	Fr. 6	Fr. 7	Fr. 8	Fr. 9	Fr. 10	Total
**Yield (mg)**	62.1	124.0	30.8	96.4	1490.0	6796.0	1487.8	1941.6	2584.6	2904.3	17,517.6
**Yield ratio (%)**	0.35	0.69	0.17	0.54	8.28	37.76	8.27	10.79	14.36	16.14	97.32
**Assay con. (μg/mL)**	1.73	3.45	0.86	2.68	41.39	188.78	41.33	53.94	71.79	80.67	

## Data Availability

Not applicable.

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
