# Peer review of "Potential of Hibiscus sabdariffa L. and Hibiscus Acid to Reverse Skin Aging"

_molecules, 2022, doi:10.3390/molecules27186076_

Round 1

Reviewer 1 Report

It is a good investigation that opens a whole field of study of how the skin ages, as well as the possible synergism of the molecules of natural products

Author Response

Point: It is a good investigation that opens a whole field of study of how the skin ages, as well as the possible synergism of the molecules of natural products. 

Response: Thank you so much for your kind encouragement! We will keep up our efforts!

Reviewer 2 Report

The present work describes the reversal of skin senescence promoted by aqueous and ethanolic extracts of Hibiscus sabdariffa L. and hibiscus acid, the latter in its chemically pure form. The authors performed a comprehensive analysis of biochemical tests to support the anti-senescence activity of the extracts of this plant, particularly the ethanolic extract, as well as hibiscus acid. Chromatographic fractionation allowed the authors to perform a bio-guided protocol to find the most active fractions. After reviewing this manuscript, I consider that it contains relevant, novel and interesting information for the specialized community, so that it can meet the quality requirements to be published in Molecules. However, a series of recommendations are listed below that must be implemented by the authors before this work can be published.

Please, NMR and mass spectra of the most representative extracts and chromatographic fractions should be included in the supplementary material.

Page 2, line 52: Delete “synthetic”

Page 3, line 66: Change “method. The dried” by “method, the dried”

Figure 1, line 71: Change “curde” by “crude”

Figure 1, line 74: Delete “and” before “LPS”

Figure 1C: Change “Collafen” by “Collagen”

Page 3, line 80: Change “Figure 1S” by “Figure S1”

Page 3, lines 91 and 103: Please revise a likely inconsistency between “18 g” and “18 mg”

Page 3, Table 1: Heading of the first column (why figure 1?)

Figure 2: In order to be consistent, change (a), (b) and (c) by their corresponding uppercase letters.

Page 4, line 121: Change “cmpound” by “compound”

Page 4, lines 125-126: Protons labeled as H-3a and H-3b do not exist in the hibiscus acid structure, they should be H-4a and H-4b.

Page 4, line 126: Change “600 MHz” by “150 MHz”

Figure 3: In order to be consistent, change (a)-(h) by their corresponding uppercase letters.

Page 6, line 141: Is the term "procession" correct?

Page 6, line 148: Change “0.8 mm” by “0.8 mM”

Figure 4: In order to be consistent, change (a)-(d) by their corresponding uppercase letters.

Page 7, line 181: Change “oxidation” by “oxidative”

Page 8, line 215: Please include the amount of dried HS used.

Page 8, line 237: Please use number 2 as a subscript in CO2

Supplementary information: It must contain the title, authors and affiliation.

Supplementary information: Figures S2-S4 are not mentioned in the manuscript; they should be briefly discussed in the manuscript.

Finally, the manuscript needs a careful revision on semantic and syntactic issues. Just one example of several, consider the last paragraph of the Discussion section (lines 195-198).

Author Response

Point 1: Please, NMR and mass spectra of the most representative extracts and chromatographic fractions should be included in the supplementary material.

Response 1: Thank you very much for your kind suggestions. We have added the mass spectra data of the representative crude extract (HSE) and the major chromatographic fractions in the supplementary information, and we are sorry for that NMR data are not available for now.

Point 2: Page 2, line 52: Delete “synthetic”

Page 3, line 66: Change “method. The dried” by “method, the dried”

Figure 1, line 71: Change “curde” by “crude”

Figure 1, line 74: Delete “and” before “LPS”

Figure 1C: Change “Collafen” by “Collagen”

Page 3, line 80: Change “Figure 1S” by “Figure S1”

Page 3, lines 91 and 103: Please revise a likely inconsistency between “18 g” and “18 mg”

Page 3, Table 1: Heading of the first column (why figure 1?)

Figure 2: In order to be consistent, change (a), (b) and (c) by their corresponding uppercase letters.

Page 4, line 121: Change “cmpound” by “compound”

Page 4, lines 125-126: Protons labeled as H-3a and H-3b do not exist in the hibiscus acid structure, they should be H-4a and H-4b.

Page 4, line 126: Change “600 MHz” by “150 MHz”

Figure 3: In order to be consistent, change (a)-(h) by their corresponding uppercase letters.

Page 6, line 141: Is the term "procession" correct?

Page 6, line 148: Change “0.8 mm” by “0.8 mM”

Figure 4: In order to be consistent, change (a)-(d) by their corresponding uppercase letters.

Page 7, line 181: Change “oxidation” by “oxidative”

Page 8, line 215: Please include the amount of dried HS used.

Page 8, line 237: Please use number 2 as a subscript in CO2

Response 2: Regarding to all the grammatical and typo errors, thank you very much for your careful review. We have corrected all the grammatical errors and labeling errors which you mentioned.

Point 3: Supplementary information: It must contain the title, authors and affiliation.

Supplementary information: Figures S2-S4 are not mentioned in the manuscript; they should be briefly discussed in the manuscript.

Response 3: Regarding to the supplementary information. Thank you for your kind reminder. The title, authors and affiliation have also been added to the supplementary information. And all supplementary materials have been briefly discussed in the manuscript.

Point 4: Finally, the manuscript needs a careful revision on semantic and syntactic issues. Just one example of several, consider the last paragraph of the Discussion section (lines 195-198).

Response 4: Thank you very much for your kind notice and reminder. We have made moderate changes on the semantic and syntactic of the manuscript, especially in the discussion section.

Reviewer 3 Report

Dear Authors 

This manuscript is well written and logically proceeded to show several physiological characteristics of aging by Hibiscus acid. 

Here are minor comments. 

1. Senecencence basically requires various characteristics such as the beta-galactosidase assay, p53/p21/p16, and flattened morphology.

Although authors showed the experiment of H2O2-induced senescence model in Fig. 3F, it was not enough to mention senescence. 

Thus, in all contents of your manuscript, senescence is not an appropriate word. Please, should replace senescence with aging simply.  

2. The discussion is too short. Authors need to write this content with more sentences.

3. References are too small. Authors need to add more references considering the quality of Molecules.  

Author Response

Point 1: Senecencence basically requires various characteristics such as the beta-galactosidase assay, p53/p21/p16, and flattened morphology. Although authors showed the experiment of H2O2-induced senescence model in Fig. 3F, it was not enough to mention senescence. Thus, in all contents of your manuscript, senescence is not an appropriate word. Please, should replace senescence with aging simply.

Response 1: Thank you very much for your careful advice. We have replaced the word “senescence” with “aging”.

Point 2: The discussion is too short. Authors need to write this content with more sentences.

Response 2: Thank you very much for your kind suggestion. We have added more rich content of discussion.

Point 3: References are too small. Authors need to add more references considering the quality of Molecules.  .

Response 3: Thank you very much for your kind suggestion, we also added more references.
